# MOTION-CATCHER: UPHOLDING MOTION AND CONTENT CONSISTENCY IN MULTI-SEQUENCE VIDEO GENERATION

## ABSTRACT

Recent developments in diffusion models have significantly advanced the field of video generation. However, technical challenges still exist in terms of spatiotemporal continuity and content consistency in long video generation. In this paper, we propose Motion-Catcher, a diffusion model-based method for multi-sequence video generation that aims to address the issues of motion inconsistency and content degradation. By incorporating a motion capture module, the model leverages optical flow information from video sequences to capture both local and global movements, enhancing the motion consistency of the videos. Furthermore, a dynamic content prior module is proposed to monitor regions prone to degradation, which helps maintain content consistency throughout the generated videos. Extensive experiments have validated that the proposed Motion-Catcher can generate videos with higher quality in terms of motion continuity and consistency. The source code and additional experimental results are available at `https://github.com/YuukiGong/Motion-Catcher`.

## 1 INTRODUCTION

In recent years, advancements in diffusion models (Ho et al., 2020; Dhariwal & Nichol, 2021; Guo et al., 2023; Zhou et al., 2022; Gupta et al., 2023; Li et al., 2018) have led to significant improvements in video generation models, particularly in terms of generation efficiency, content quality, and consistency. The open-sourcing of many video generation models, pre-trained on large datasets (Chen et al., 2024b; Blattmann et al., 2023b; Rombach et al., 2022; Jiang et al., 2024; Zhang et al., 2023a;c; Chen et al., 2024a; Wang et al., 2023), has drawn more attention from researchers. After substantial training, the generated videos show considerable improvements in stability and diversity. Although current diffusion models have achieved good results in video generation, they still face certain limitations when generating longer videos, such as motion inconsistency and content degradation. Given that most video generation models are trained within a limited frame count range, the network struggles to maintain consistency over longer durations. Additionally, hardware limitations often constrain training and inference on longer video streams due to memory restrictions. Considering the limited hardware, videos can be generated sequentially using an autoregressive approach to mitigate content degradation. However, final videos assembled from these sequences often exhibit varying speeds and directions of motion in each sequence, leading to temporal inconsistencies in the final output. Fine-tuning methods have been developed in the image generation domain (Hu et al., 2021; Ruiz et al., 2023; Gal et al., 2022; Zhang et al., 2023b) to stabilize the production of desired content and have been successfully adapted to video generation, demonstrating strong performance in short video contexts. However, these methods encounter significant challenges in maintaining motion and content consistency in longer videos. This difficulty primarily stems from the fragmentary nature of autoregressive generation, which frequently results in abrupt transitions. Thus, resolving the issues of motion inconsistency and content degradation in multi-sequence video generation still remains underexplored comprehensively.

Existing methods to control video content generation can be classified into two directions: 1) Fine-tuning the temporal and spatial layers in the network (Wu et al., 2023) or adding Low-Rank Adaptation (Xu et al., 2024) to achieve motion and content control; 2) Training additional auxiliary networks (Hu, 2024; Chen et al., 2023c; Ling et al., 2024; Materzynska et al., 2023) to compensate

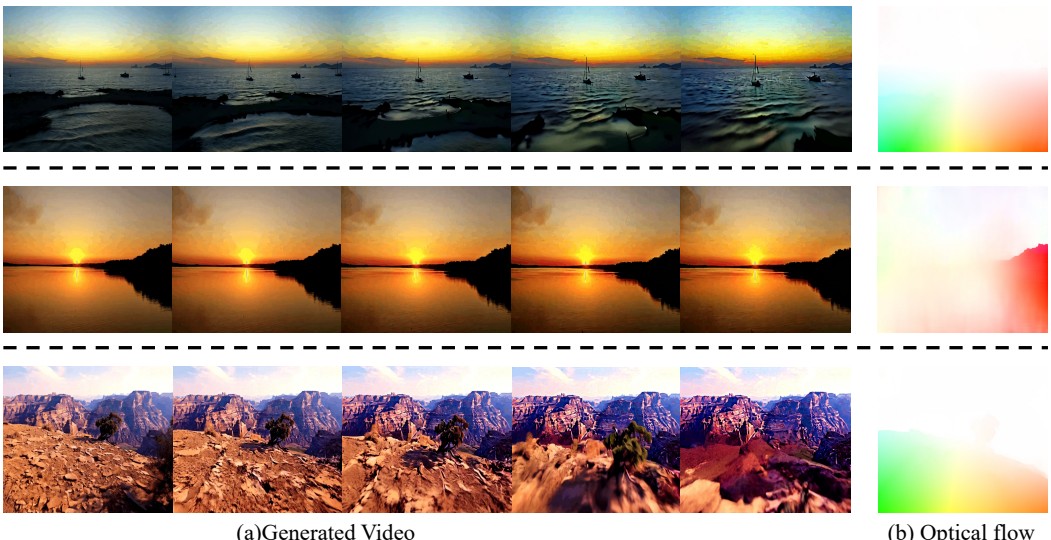

(a)Generated Video                (b) Optical flow

Figure 1: **Examples of the generated videos.** Motion-Catcher maintains strong consistency across multiple video sequences. The images on the left represent a subset of the final video sequence, while the right shows the averaged optical flow of the motion in the video. The original resolution is 1024×576, but it has been rescaled here for display.

for the outputs of the original model, or setting control factors in the network for precise control. Despite these efforts, the autoregressive generation of long videos continues to exhibit inconsistent camera movements and variations in the main subjects. Recent works, including those by Chen et al. (2023c) and Chen et al. (2023b), have introduced predefined motion signals—such as depth images and edge information—to direct video content. While these inputs effectively constrain main subjects and backgrounds, they often only preserve edges or shapes over long durations, leading to content degradation.

To tackle the challenges of inconsistent motion and the degradation of primary subjects in autoregressive multi-sequence video generation, we propose Motion-Catcher, a novel method designed to regulate the diffusion model for the stable generation of multi-sequence video content. This method effectively captures both motion dynamics and the integrity of the main subject content, as illustrated in Figure 1, which presents examples of the generated videos. Specifically, the model first generates an initial sequence based on given content and then captures the motion information of the scene and camera, as well as the content information of moving objects within the sequence, to guide the generation of subsequent sequences. Finally, the generated video can be obtained through the concatenation of multiple continuous video sequences.

Motion-Catcher employs a motion capture module that utilizes local and global optical flow information from video sequences which indicates the movement of objects and the camera. This motion information is then integrated into the generation process of the following video sequence to provide motion compensation, enhancing the consistency of video motion. However, research has found that merely introducing motion information does not fully resolve the degradation of the video content's main subject that occurs through continuous iterations, specifically manifested by changes in the appearance and shape of objects within the video. To alleviate this degradation, we also introduce a dynamic content prior module. This module uses global absolute optical flow and calculations between image frames to identify areas prone to degradation during the generation process, thereby incorporating prior information about dynamic objects to maintain content consistency. In generating the initial sequence, zero optical flow information is used to produce videos with stable motion. The subsequent video sequence is generated using a conditional diffusion process. This process incorporates the final frame of the current video sequence, along with the captured motion information and content priors, to ensure consistency.

Experimental results have demonstrated that by incorporating the motion capture module and dynamic content prior module, the proposed Motion-Catcher successfully maintains both video motion and content consistency while generating each sequence. In the AIGCBench benchmark (Fan et al., 2024), the proposed method shows motion consistency and content preservation improvements, sur-

passing current mainstream methods. Furthermore, our proposed method can be easily integrated into other diffusion-based video generation methods to enhance visual consistency. Additionally, this method effectively avoids fitting to the limited appearances of reference videos, maintaining motion across different appearances. Our contributions are summarized as follows:

- We present an efficient method for generating multi-sequence videos, designing a motion capture module that generates motion-consistent video sequences autoregressively and maintaining motion consistency in complete video generation.

- To mitigate content degradation over time in video generation, a dynamic content prior module is proposed, combining global absolute optical flow and capturing the shapes of dynamic objects to maintain the content consistency in the generated videos effectively.

- The proposed motion capture module and dynamic content prior module can be applied as plug-and-play components in different video diffusion generation models. Experimental results have demonstrated that Motion-Catcher produces long videos with improved continuity and stability.

## 2 RELATED WORK

**Video Diffusion Models.** Recent work on video generation has predominantly utilized diffusion models, extending the concept of diffusion to higher dimensions and more complex video data, achieving progressive generation from random noise to high-quality video sequences. Leveraging the success of diffusion models in image generation and guided by text or image conditions, video diffusion models pre-trained on large datasets have been capable of producing high-quality video sequences while maintaining diversity and stability. Among these, unlike operating directly in the pixel space, Yu et al. (2023) operate in latent space, using latent variables to represent high-level features in video data that encapsulate dynamic changes, object relations, and scene structures. He et al. (2022) have introduced lightweight video diffusion models that utilize a low-dimensional latent space to generate smooth video results within a limited computational budget, demonstrating significant potential and broad application prospects. This improves the quality and diversity of generated videos and optimizes computational efficiency. Most related work incorporates various forms of temporal mixing layers into the training architecture to learn connections between video sequences, with researchers like Ge et al. (2023) focusing on temporally correlated noise to enhance temporal consistency and simplify the learning task. Blattmann et al. (2023b) introduce temporal convolution and attention layers after each spatial convolution and attention layer to more broadly capture the motion relationships between video frames. Recent video generation models (Blattmann et al., 2023a) have shown promising results by screening and filtering large datasets during the training and fine-tuning stages of video diffusion generation models, further fueling users' enthusiasm for creating imaginative videos.

**Model Fine-tuning for Customization.** Customizing pre-trained large foundational models can better adapt to downstream tasks and user preferences while maintaining strong generative knowledge without the need to train from scratch. Several mature customization methods have been developed for generating images with specific subjects and styles in the image generation domain and have been successfully transferred to video generation. In video generation, fine-tuning methods must consider not only style and content consistency but also the consistency of inter-frame motion. Methods such as Dreambooth (Wei et al., 2024) and Lora (Hu et al., 2021) have demonstrated effective results in image fine-tuning and have successfully adapted to video generation, producing videos that reflect specific styles based on reference sequences. However, these approaches are overly dependent on the fitted videos and exhibit challenges in capturing motion information effectively, resulting in difficulties in consistently generating long video content.

**Controllable Video Generation.** Many video generation methods incorporate different motion control mechanisms to generate videos with specific motions. Techniques such as Moonshot (Zhang et al., 2024) and MotionClone (Ling et al., 2024) capture motion from a source video and apply these movements to animate static images. These techniques rely heavily on video material and still face challenges in achieving high levels of detail and precise manipulation. In contrast, manipulable video synthesis technology (Yin et al., 2023; Wu et al., 2024) operates by learning a set of input instructions, allowing users to interactively adjust the positions of objects within a scene and produce smooth, continuous video sequences. However, these learned instructions concerning motion are often simple and coarse-grained. The Control a Video method (Chen et al., 2023c) trains additional

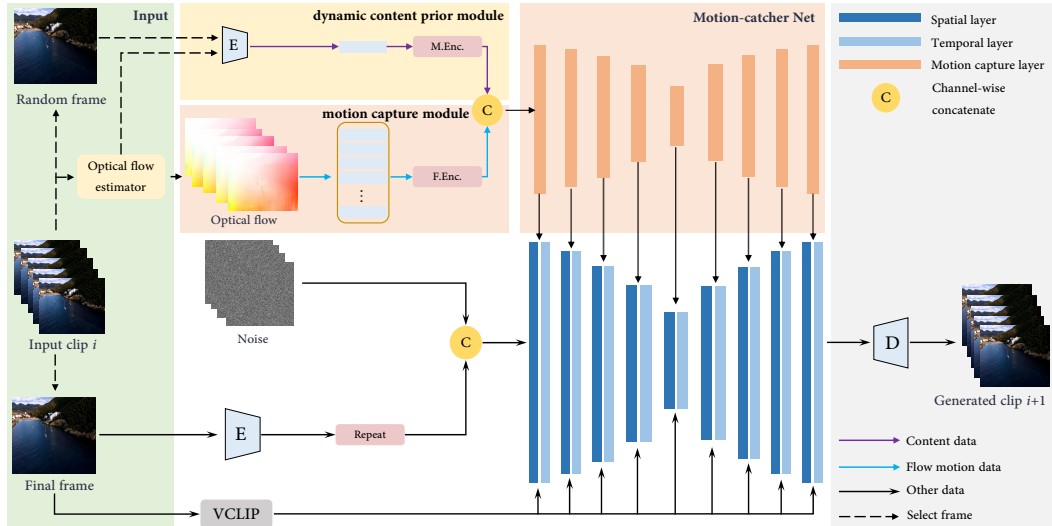

Figure 2: **The framework of Motion-Catcher.** The initial sequence begins with a reference image and zero optical flow to generate the first video sequence. Subsequent sequences use the optical flow and the final frame of the previous video sequence for guidance, ensuring motion consistency. All sequences are combined to form a final video with consistent visual content and motion.

branch networks that accept conditional signals to align the generated video with these conditions. The control signals are usually extracted from a reference video, adjusting the appearance and motion of the generated video, leading to outcomes influenced by the appearance and motion present in the reference video. When it comes to maintaining continuous video sequences of motion states and main subject content, the lack of additional reference video input makes the direct application of these methods challenging.

## 3 METHOD

The proposed Motion-Catcher captures both motion trends and dynamic content from the initial scene and perpetuates them throughout the generation of subsequent video sequences. As depicted in Figure 2, the video generation process commences with the creation of an initial video sequence followed by subsequent sequences. Initially, a reference image and zero optical flow are input into the module. The reference image dictates the content and semantics of the video, while the optical flow is fed into the motion capture module to extract motion information. In scenarios with zero optical flow input, the module struggles to capture precise motion information, resulting in a sequence characterized by random movements that inadvertently enhance output diversity. The generation of subsequent video sequences leverages the motion and content information from earlier sequences. This method ensures that each sequence aligns with the motion patterns and semantic context of preceding sequences, thereby effectively preserving consistency in motion and content across the video.

In the proposed method, two novel key modules are designed, which are the motion capture module and the dynamic content prior module. The motion capture module receives image frames and a series of optical flow sequences as inputs, which are utilized to extract the motion information crucial for video generation. The dynamic content prior module uses accumulated absolute optical flow to identify regions within the video frames susceptible to degradation. A mask is applied to these frames before they are processed by the module. This preprocessing step allows for targeted adjustments in subsequent outputs, aimed at preserving the shape and semantic integrity of objects within the video frames, thereby ensuring content consistency of the output video sequences.

### 3.1 MOTION CAPTURE MODULE

The relationship between motion and optical flow can be conceptualized as a mapping, where motion refers to the physical displacement of objects in the real world, and optical flow represents the

projection of this motion onto the imaging plane. Based on this relationship, we design a motion capture module, aiming to derive the video's motion state from optical flow while maintaining motion consistency across newly generated videos. This process is divided into two stages: the generation of the initial video sequence and the subsequent sequences. In the initial video generation stage, it is essential to establish the overall visual content, semantic information, and the direction and speed of motion, which serve as prior information for subsequent sequences. For visual content, the model takes a reference image as input, which is then embedded into a latent space and replicated $N$ times, where $N$ corresponds to the length of the video sequence being generated. To initiate the diffusion model, random Gaussian noise is concatenated along the channel dimension, providing an initial distribution. In the generation of subsequent video sequences, the final frame of the previous sequence is extracted as the starting image and processed in the similar manner.

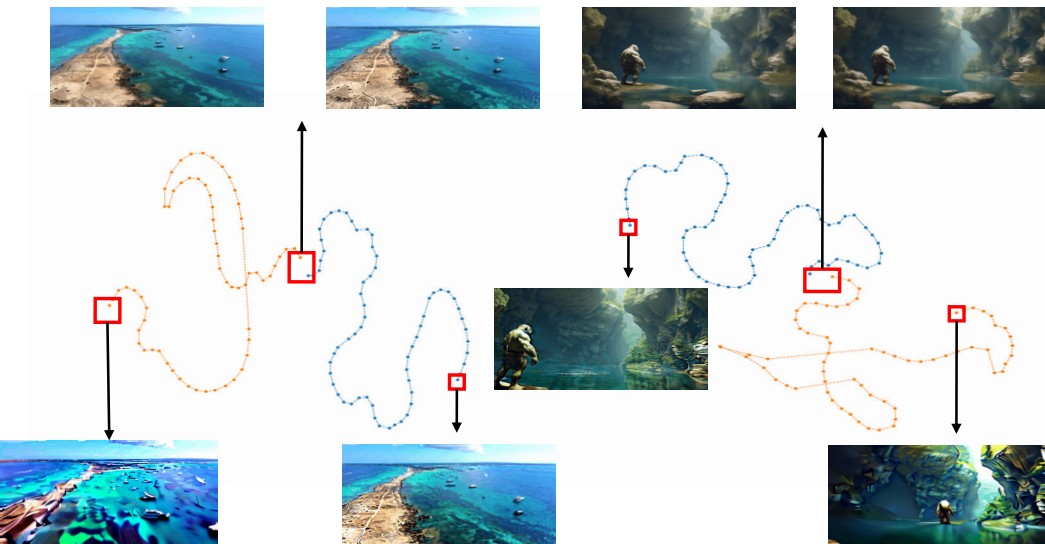

Figure 3: **Latent space distribution of generated results.** This t-SNE plot visualizes the latent space of video sequences, with blue representing sequences processed through the motion capture module and orange indicating sequences generated directly from original images. The analysis shows that the inclusion of the motion capture module yields a smoother distribution in the latent space, reflecting enhanced consistency in the video sequences.

For the initial video sequence generation, the model processes zero optical flow as input to describe the motion information. This approach enables the model to autonomously generate motion states that are compatible with the given image space, ensuring both the rationality of motion and the preservation of diversity in motion states. During the training phase, the input optical flow is set to zero with a specified probability $\alpha$ to mirror the condition under which the initial sequence receives no optical flow input. In the process of generating subsequent video sequences, dense optical flow estimation is performed between each pair of adjacent frames in the currently fitted video sequence, yielding a series $S = \{S_{1-2}, S_{2-3}, \ldots, S_{(n-1)-n}\}$ that represents the local motion states. Additionally, to provide comprehensive motion information, it is crucial to estimate the optical flow from the first to the last frame of the image series, $S_{1-n}$, and concatenate this series along the temporal dimension. These optical flow data are then fed into the Flow Motion Encoder (F.Enc), which consists of a combination of four 3D convolutional layers, four temporal attention layers, and four downsampling convolutional layers. We further enhance the model by incorporating interactions that extend self-attention across all optical flow sequences:

$$\text{Attention}(\boldsymbol{Q}, \boldsymbol{K}, \boldsymbol{V}) = \text{Softmax}(\frac{\boldsymbol{Q}\boldsymbol{K}^T}{\sqrt{d}}) \cdot \boldsymbol{V}, \text{ where } \boldsymbol{Q} = \boldsymbol{W}^Q\boldsymbol{s}, \ \boldsymbol{K} = \boldsymbol{W}^K\boldsymbol{s}, \ \boldsymbol{V} = \boldsymbol{W}^V\boldsymbol{s}, \quad (1)$$

where $\boldsymbol{s} = \{\boldsymbol{s}^i\}_{i=0}^{N-1}$ denotes all lateoptical flow, while $\boldsymbol{W}^Q$, $\boldsymbol{W}^K$, and $\boldsymbol{W}^V$ project $\boldsymbol{s}$ into query, key, and value, respectively.

Figure 3 demonstrates that continuous video sequences exhibit smooth trajectories in latent space, indicating consistent motion handling. Conversely, if the final frame of a video is used to autoregres-

sively generate subsequent videos, which are then concatenated into a complete video, the resulting motion trajectory appears disordered. In pixel space, this manifests as inconsistent motion states, highlighting the crucial role of this module in providing motion compensation to ensure each video sequence consistently maintains its motion state.

## 3.2 DYNAMIC CONTENT PRIOR MODULE

The proposed Motion-Catcher method utilizes images to provide reference content and semantic information for the generated videos. This approach, which diverges from text-conditioned video generation models, uses images as conditional inputs to directly influence the distribution of the output. During generation, the input image is merged with Gaussian noise in latent space, forming the initial input distribution for the diffusion model. This method ensures continuity in content and semantics across the generated video sequences. However, without additional constraints, the model struggles to maintain long-term global content consistency. Over time, the background and main subjects in the generated videos undergo significant alterations, eventually becoming unrecognizable, as depicted in Figure 4. Studies have indicated that dynamic scenes tend to progressively lose their original appearance and contour shape under sustained motion, resulting in noticeable degradation.

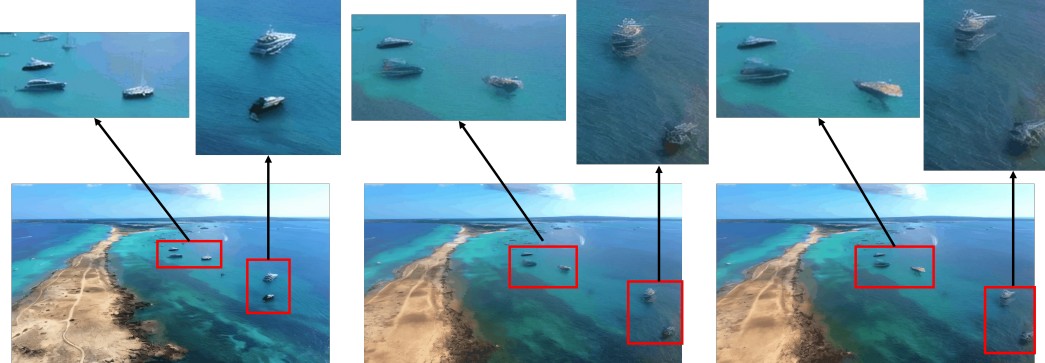

Figure 4: **Changes in the appearance of objects in motion.** As time goes by, the main content of the video will change during the generation.

Based on the above analysis, we propose a dynamic content prior module that utilizes global and local motion magnitudes derived from optical flows. The global absolute optical flow $S_{all}$ and global average motion magnitude $S_{mean}$ can be computed as:

$$S_{all} = \sum_{i=1}^{T} |F_i|, \quad S_{mean} = \frac{1}{N} \sum_{x,y} S_{all}(x,y), \quad I'(x,y) = \begin{cases} I(x,y) & \text{if } S_{all}(x,y) > S_{mean}, \\ 0 & \text{otherwise.} \end{cases} \quad (2)$$

where $T$ is the total number of frames, $|F_i|$ is the absolute value of the optical flow matrix for frame $i$, $N$ is the total number of pixels, $S_{all}(x,y)$ represents the global absolute optical flow at pixel position $(x,y)$, $S_{mean}$ is the global average motion magnitude, and $I'(x,y)$ is the filtered image based on motion magnitude.

The obscured appearances from the videos are transformed into latent space and subsequently input into a mask encoder, designated as M.Enc, for content feature extraction. This encoder comprises four naive convolutional layers and self-attention layers, adeptly extracting pertinent features from obscured images. During the generation of the initial video sequence, which typically lacks prior informational content, the dynamic content prior module employs entirely obscured images as inputs. Throughout the training phase, this module is deliberately fed obscured images with a set probability $\beta$, mirroring the conditions during the inference process of the initial video sequence generation. This approach ensures the model's training aligns closely with its operational environment. The features extracted by the dynamic content prior module are then integrated with those from the motion capture module. As illustrated in Figure 5(a), the absence of the motion capture module results in inconsistent camera movements, where the camera initially moves left and then abruptly changes direction, leading to increasingly chaotic and blurred sequences. In contrast, the integration of the

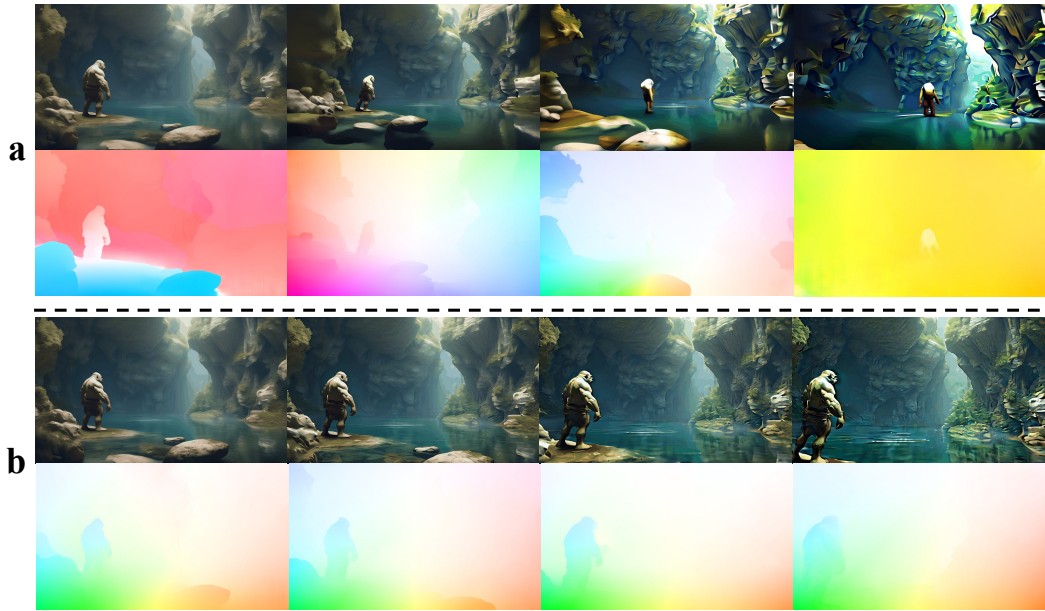

Figure 5: **Sampling and Optical Flow in Generated Video Sequences.** Panel (a) displays sampled frames from multiple sequences without the motion capture module, while panel (b) displays sampled frames from multiple sequences with the motion capture module.

motion capture module, as shown in Figure 5(b), significantly enhances the consistency and clarity of the generated video sequences. This module not only maintains the direction and frequency of object and camera movement but also effectively acts as content compensation for the video output. The improvement is particularly pronounced in the optical flow charts, where each slice consistently shows different scenes and subject motions, demonstrating the method's effectiveness in enhancing the overall quality and coherence of the video.

Finally, the motion features are concatenated with content features along the channel dimension before being fed into the Motion-catcher Net. This integration ensures that the generated subsequent video sequences preserve the same motion state as the previous sequences. The spatiotemporal self-attention mechanism within the net adeptly captures the rate of motion across the optical flow series, securing consistency in motion direction and frequency. This allows for fine-grained modeling of motion dynamics. During the model inference phase, we adopt the classifier-free guidance method for diffusion (Ho & Salimans, 2022) to guide the iterative refinement process of the diffusion model. This method reduces the need for additional supervisory signals and enhances the diversity and efficiency of generation, allowing the generated content to vary according to different conditions or directives while maintaining high-quality output:

$$D^w(\mathbf{x}; \sigma, \mathbf{c}) = wD(\mathbf{x}; \sigma, \mathbf{c}) - (w - 1)D(\mathbf{x}; \sigma) \tag{3}$$

where $w \geq 0$ is the guidance strength, $D(x; \sigma, c)$ is the conditional diffusion model output, $D(x; \sigma)$ is the unconditional diffusion model output, $x$ represents the input data, $\sigma$ is the noise level, and $c$ denotes conditioning variables. In generating video sequences, too little guidance might lead to inconsistency with the conditional framework, whereas too much guidance might result in oversaturation. Therefore, a linearly increasing guidance value was used to maintain a stable generation state.

## 4 EXPERIMENTS

### 4.1 IMPLEMENTATION DETAILS

**Dataset Setup.** In this paper, we train the Motion-Catcher on 20k high-quality video clips sourced from the internet. These clips have stable motion states, high video resolution, and significant perspective changes. We test the Motion-Catcher on 2k images employed for image-to-video generation from AIGC (Fan et al., 2024).

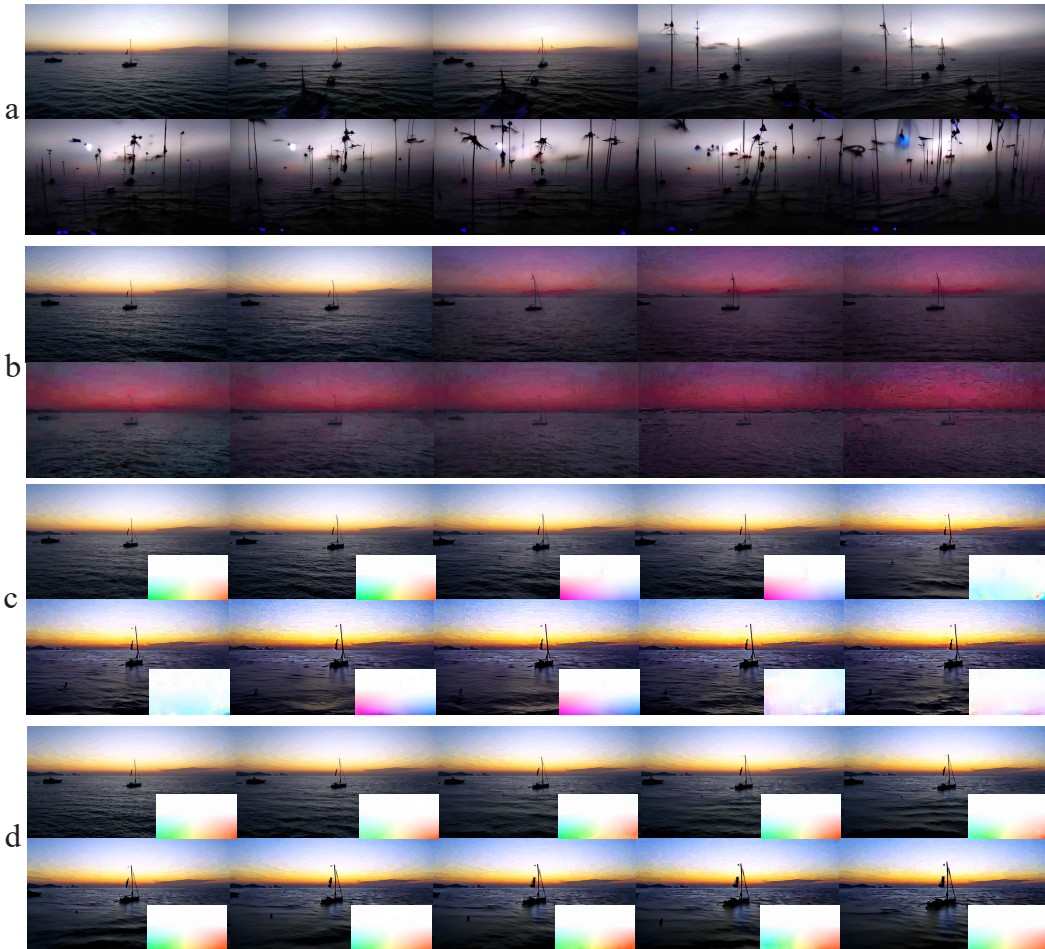

Figure 6: **Consistency comparison.** We select a fixed image as input and autonomously generate five video sequences in an autoregressive manner, sampling linearly from the output. The panels are labeled: (a) VideoCrafter, (b) I2VGen-XL, (c) SVD, and (d) Our Motion-Catcher. In panels (c) and (d), optical flow maps in the bottom-right depict motion between frames. Notably, the optical flow in (d) consistently remains within a specific range, indicating stable motion, while (c) shows varying motion directions.

**Model Training.** For a given video, we select a sequence of 28 consecutive frames for training, with the first 14 frames used for motion information extraction and the remaining 14 for fitting the video generation results. For the base model, we use pre-trained parameters from SVD (Blattmann et al., 2023a) to initialize U-Net's weights, enabling the Motion-Catcher to generate short video sequences. During the inference phase, for generating initial video sequences, the model receives zero optical flow and entirely obscured images as inputs for motion information description and dynamic content priors. During training, we set $\alpha$ and $\beta$ at 0.1 to transition the input optical flow to zero and dynamic content priors to complete obscuration, aligning with the process of initial sequence reception of zero optical flow and entirely obscured image inputs. We optimize the proposed network for 30k steps using AdamW (Loshchilov, 2017) with a batch of size 32, and the learning rate is set to 2e-5. In the optical flow estimation, we use the latest deep optical flow estimation method SEA-RAFT (Wang et al., 2024), a new RAFT (Teed & Deng, 2020)variant incorporating architectural changes for enhanced efficiency and accuracy.

### 4.2 MAIN RESULTS

#### 4.2.1 QUALITATIVE ANALYSIS

We compare our method with mainstream image-to-video generation models (Chen et al., 2023a; Zhang et al., 2023c; Blattmann et al., 2023a). For each model, videos are generated according

Table 1: **compares Motion-Catcher against other methods.** The metrics presented include MSE, SSIM, Image-GenVideo Clip, Flow-Square-Mean, GenVideo Clip, and Frame Count.

| Method | Control-video Alignment | | | Motion Effects | Temporal Consistency | |
|---|---|---|---|---|---|---|
| | MSE(First) ↓ | SSIM(First) ↑ | Image-GenVideo Clip ↑ | Flow-Square-Mean | GenVideo Clip ↑ | Frame Count ↑ |
| Video Crafter | 3929.65 | 0.3 | 0.83 | 1.24 | 0.98 | 16 |
| I2VGEN-XL | 4491.9 | 0.354 | 0.832 | 1.8 | 0.971 | 32 |
| SVD | 640.75 | 0.612 | 0.919 | 2.52 | 0.974 | 25 |
| **ours** | **156.23** | **0.762** | **0.947** | 1.37 | **0.994** | **70** |

to their trained size and frame numbers and are concatenated in an autoregressive manner to form multi-sequence videos. In Figure . 6, given the same input images, VideoCrafter generates additional content and changes the visual style in subsequent video sequences. I2VGen-XL, while maintaining overall content consistency, shows a shift in coloration and a progressively blurring central image until it disappears. In SVD, inconsistencies in motion are observed, with discontinuities in the camera and scene across sequences. Compared with the above methods, our proposed Motion-Catcher method performs best.

### 4.2.2 QUANTITATIVE ANALYSIS

we use the evaluation metrics proposed in AIGCBench, which has established a comprehensive and standardized system of assessment criteria designed to meet mainstream video generation tasks. The generated videos are evaluated from four dimensions: control-video alignment, motion effects, temporal consistency, and video quality. More importantly, these metrics operate independently of reference videos in image-to-video generation, ensuring a thorough assessment strategy. We compare these methods to other open-source models for image-to-video tasks, including VideoCrafter, I2VGen-XL, and Stable Video Diffusion.

In terms of control-video alignment, AIGCBech utilizes image fidelity metrics to evaluate how similar the generated video frames are to the initial image inputted into the model (particularly the first frame). To assess fidelity, Mean Squared Error (MSE) and Structural Similarity Index (SSIM) are employed to gauge the retention of the first frame. For overall video frames, the image-to-generated video CLIP similarity is calculated for each frame. These evaluations are represented by the metrics MSE (First), SSIM (First), and Image-Generated Video CLIP. Regarding motion effects, the primary evaluation concerns whether the motion in the generated videos is significant and reasonable. The motion effect's magnitude is gauged by calculating the optical flow scores between adjacent frames, with the average value representing the extent of motion. The squared mean of predicted values between adjacent frames describes the video's dynamic motion, where higher values indicate stronger motion effects. In accordance with the strategy outlined in AIGCBench, we establish a threshold: the Flow-Square-Mean must be less than 10 to filter out undesirable scenarios. In the aspect of temporal consistency, we mainly measure whether the generated video frames are consistent and coherent. The CLIP similarity between every two adjacent frames in the generated video is calculated, with the average value serving as the metric for the video's temporal consistency, denoted as GenVideo Clip (Adjacent frames).

The final video is merged from five sequences, each consisting of 14 frames. As shown in Table 1, our method achieves optimal levels in MSE and SSIM for the first generated frame compared to the given content, indicating that our approach preserves more information from the initial image. Despite a higher frame count, the Flow-Square-Mean value remains elevated, demonstrating significant motion in each frame and richer motion information with increasing frame numbers. Additionally, in the Temporal Consistency column, our results surpass other methods, illustrating that our model can maintain the camera's motion and content across each video sequence and overall semantic consistency. It shows our method's capability to sustain intrinsic motion states during complex transformations while managing video semantic coherence. By integrating motion information and dynamic content priors, our approach captures and perpetuates the initial scene's motion trends and dynamics, ensuring natural and physically plausible motion trajectories even in video sequences involving significant action changes or scene transitions.

### 4.3 ABLATION STUDIES

Figure. 7 shows the content of frames at different moments during video generation. When the dynamic content prior is applied, the main subjects in the scenes are maintained while suppressing

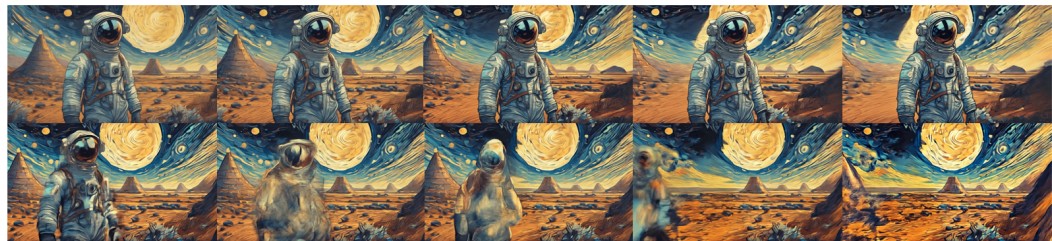

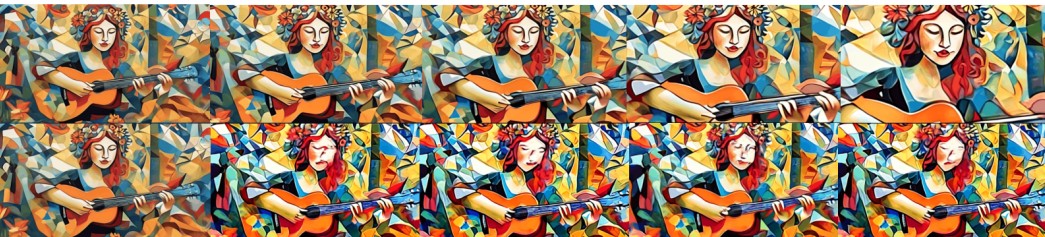

Figure 7: **Comparison of Video Frame Generation with and without the Dynamic Content Prior Module.** The top row illustrates frames enhanced by the dynamic content prior module, while the bottom row displays frames generated without it, highlighting the module's impact on content consistency and detail in video sequences.

Table 2: Comparison of different configurations against the baseline

| Method | Control-video Alignment | | | Motion Effects | Temporal Consistency | |
| --- | --- | --- | --- | --- | --- | --- |
| | MSE(First) ↓ | SSIM(First) ↑ | Image-GenVideo Clip ↑ | Flow-Square-Mean | GenVideo Clip ↑ | Frame Count ↑ |
| Baseline | 696.35 | 0.554 | 0.904 | 1.24 | 0.986 | 70 |
| Baseline + Motion | 302.67 | 0.697 | 0.93 | 3.62 | 0.991 | 70 |
| Baseline + Motion + Content | **156.23** | **0.7623** | **0.947** | 1.37 | **0.994** | 70 |

changes in visual style. These findings indicate that Motion-Catcher can preserve scenes' content over longer durations, including the shapes of subjects and their surrounding environments.

We also conduct a quantitative analysis of the motion capture module and the dynamic content prior module, evaluating the generated videos across key dimensions: control-video alignment, motion effects, temporal consistency, and video quality. The results are summarized in Table 2. Ablation studies reveal that the exclusion of the motion capture module significantly downgrades motion coherence. Furthermore, the application of the dynamic content prior leads to improved semantic coherence, resulting in an overall superior performance. This combination effectively mitigates content degradation during extended video generation, thereby enhancing visual coherence in the produced videos.

## 5    CONCLUSION

In this paper, we present a diffusion model based method for multi-sequence video generation. It can capture and sustain the motion trends and dynamics of the initial scene throughout the video frame generation process. This method ensures that even sequences with significant action changes or scene transitions exhibit natural, fluid, and physically plausible motion trajectories. By utilizing the proposed motion capture module to regulate the direction and frequency of motion, alongside integrating a dynamic content prior module to enhance content consistency, our method yields better coherent and visually reasonable videos. However, there are still shortcomings in the diffusion model based methods. Current image-to-video diffusion model approaches often experience style drift over extended temporal dimensions, leading to gradual alterations in visual style. Future research is needed to explore strategies for enhancing style stability and controllability in video generation models, thereby addressing these ongoing challenges.

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
