# OpenReview forum: "Motion-Catcher: Upholding Motion and Content Consistency in Multi-Sequence Video Generation"
_ICLR.cc/2025/Conference — ICLR 2025 Conference Withdrawn Submission_

### Official Review · Reviewer_JULg · 2024-11-01

**Soundness:** 2
**Presentation:** 2
**Contribution:** 2
**Rating:** 3
**Confidence:** 5

**Summary:**

This paper presents a method, named Motion-Catcher, for sequence-wise long video generation. The claimed main contributions are a motion capture module, which intends to enhance the motion consistency, and a dynamic content prior module, which intends to avoid quality degradation at certain regions. The model is trained with 20K high-quality video clips and evaluated on AIGCBench. System-level comparison shows that the proposed Motion-Catcher has a better performance than Video Crafter, I2VGen-XL, and SVD in terms of control-video alignment and temporal consistency. Selected results are provided on a github link.

**Strengths:**

This paper addresses a relevant and important problem, which is the appearance and motion consistency in long video generation. This work provides a solution that partially addresses the problem and may bring inspiration to other work.

**Weaknesses:**

The main weakness is that the model developed does not generate meaningful and fine-grained motion. All the examples shown in the paper and in the results page only contain nearly static objects with some camera motion, such as pan, zoom in, or zoom out. In fact, fine-grained motion, such as people walking or sea waving, cannot be characterized by smooth optical flow. Therefore, the method has intrinsic flaws.
Both qualitative and quantitative results are not satisfactory. In quantitative results, the authors adopt the evaluation metrics proposed in AIGCBench, but it is not clear why only a subset of the metrics are adopted.

**Questions:**

What are the quantitative results for video quality evaluation? Would you please report all the metrics proposed in AIGCBench?
While FVD on UCF101 is not an ideal metric for synthesized video evaluation, it does provide some hints about how the generated motion aligned with motions in natural videos. The authors may take the first frame in each video as the reference frame to generate subsequent clips, and evaluate the FVD for the first, second and later sequences generated by Motion-Catcher.

---

### Official Review · Reviewer_LRWa · 2024-11-05

**Soundness:** 3
**Presentation:** 1
**Contribution:** 2
**Rating:** 5
**Confidence:** 5

**Summary:**

This paper introduces Motion-catcher, a method that generates videos with temporally consistent motion and content. Motion-catcher consists of a motion capture module that autoregressively generates motion-consistent long video and a dynamic content prior module that takes the global motion and object information into account for content consistency. The proposed module can be readily plugged into various diffusion-based video generation pipelines. Experiments validate the effectiveness of Motion-catcher on several datasets.

**Strengths:**

1. Using optical flow to capture the motion information of a video is intuitive and the authors manage to make it work for the proposed method.

1. Autoregressive generation of video clips seems promising for generating long videos.

1. Experiments demonstrate the effectiveness of the proposed method.

**Weaknesses:**

1. The presentation of this paper could be improved.
- I am a bit confused about Figure 1. A clean average optical flow does not indicate temporal consistency but may indicate that the generated motion is very smooth (smoothness does not suggest high quality). The same thing happens to Figure 3, I do not see why smoother motion indicates better motion consistency and more visually appealing generation results -- highly dynamic motions (e.g., human dancing, shaking cameras, etc) could also be desirable during image generation.
- In line 208, the authors wrote "A mask is applied..." What does this mask look like? It would be better to show this mask in Figure 2 or explain it using text.
- In Figure 2, how can the same encoder E only take the final frame as input while also taking the random frame and the output of the optical flow estimator as input in the dynamic content prior module?
- In line 266, what does lateoptical flow mean? Is it a typo?
- What do the videos in the supp want to demonstrate? What is the difference between before1.mp4 and after1.mp4? Before and after what? If the before1.mp4 is the result without using the proposed method, I highly doubt the result, as a video generation model like SVD could generate better videos if tuned properly.
- What is the resolution of the generated video? Figure 1 suggests that the video is 1024×576; however, in the supplementary material, videos are 512 × 288.

2. The technical novelty of this paper is weak. Using optical flow as the motion feature to capture temporal dynamics is straightforward, and the proposed motion capture module does not provide any surprises. The authors should at least give a thorough analysis of the design choice.

3. The generated videos present various artifacts (e.g., the foreground and background of after1.mp4 move separately without any dynamic motion, after4.mp4 only shows a zoom-in effect without foreground motion). I am expecting more visually appealing results, for example, foreground humans with more dramatic motion and the objects in the pictures are all moving instead of staying static. Moreover, the proposed method could theoretically generate minute-long videos, I would like to see such results.

4. The authors stated that "The proposed motion capture module and dynamic content prior module can be applied as plug-and-play components in different video diffusion generation models." However, I did not find the experiments on different video generation models. It seems that the authors only conduct experiments on SVD.

5. The input to the dynamic content prior module is randomly selected, is there a better way to select the optimal frame? Or using multiple frames as input? Does this randomness affect the performance? If an object is missing in the randomly selected frame but present in other frames, it may result in content inconsistency regarding the missing object.

**Questions:**

The inadequate representation of this paper makes it hard to understand the merits of this paper. The technical novelty of this paper should also be improved. Stronger experiments are required. A deeper investigation of incorporating motion and content features should be conducted. Considering the current state of this paper, it may not be ready for publication at ICRL.

---

### Official Review · Reviewer_m11L · 2024-11-08

**Soundness:** 3
**Presentation:** 2
**Contribution:** 2
**Rating:** 5
**Confidence:** 4

**Summary:**

The paper proposes a model called Motion-Catcher. This model is a diffusion model designed to enhance motion and content consistency in multi-sequence video generation. The model addresses common issues in long video generation, such as motion inconsistency and content degradation, by introducing two main components: a motion capture module that leverages optical flow information for enhanced motion continuity, and a dynamic content prior module to mitigate content degradation over time. Experimental results demonstrate that Motion-Catcher significantly improves video quality, stability, and coherence compared to other models, with applicability as a plug-and-play enhancement for other video diffusion models.

**Strengths:**

1. The proposed method introduces solutions to longstanding issues in video generation, namely motion inconsistency and content degradation in long video sequences.
2. The motion capture and dynamic content prior modules are well-designed, providing complementary functions to achieve both motion and content stability across video sequences.
3. The paper includes qualitative and quantitative evaluations, as well as ablation studies, which validate the model's effectiveness over existing methods.
4. The model is versatile, with components designed to be integrated into other video diffusion models, potentially broadening its applicability.
5. Motion-Catcher consistently outperforms baseline models on standard metrics, such as MSE, SSIM, and temporal consistency, indicating its robustness.

**Weaknesses:**

1. The paper writing needs to be improved. It would be better to pay more attention to the meaning of this paper, including task definition, motivation of experiment design, notations in methodology, etc.
2. This model needs some well-designed user studies since the motion consistency should be evaluated by humans.
3. The model is designed to modify the generated video clip into a more consistent one. Please describe why not design modules to improve the motion consistency for the generated video clip the first time. I believe it is a hard task to find the anti-fact details and fix them.
4. The dataset, AIGC (Fan et al., 2024), used in this paper is not well-known. It would be better to use some widely used datasets (e.g., MSCOCO, LAION-2B, UCF-101, Cityscapes).
5. It would be better if the discussion in related works included methods for video generation with optical flow (e.g., [1r, 2r, 3r]).
6. Minor:
(1) Line 432: "Table 1: compares Motion-Catcher against ..." -> "Comparison of Motion-Catcher against ..."
(2) Line 441, Line 485: "Figure. 6" -> "Figure 6". The latex code should be "Figure~\\\ref{xxx}."
(3) Line 449: "we" -> "We"


[1r] Liang, Feng, et al. "Flowvid: Taming imperfect optical flows for consistent video-to-video synthesis." Proceedings of the IEEE/CVF Conference on Computer Vision and Pattern Recognition. 2024.

[2r] Liang, Jingyun, et al. "MoVideo: Motion-Aware Video Generation with Diffusion Model." European Conference on Computer Vision. Springer, Cham, 2024.

[3r] Ni, Haomiao, et al. "Conditional image-to-video generation with latent flow diffusion models." Proceedings of the IEEE/CVF conference on computer vision and pattern recognition. 2023.

**Questions:**

Please address my concerns above. Thank you!

---

### Official Review · Reviewer_g61E · 2024-11-08

**Soundness:** 2
**Presentation:** 2
**Contribution:** 2
**Rating:** 3
**Confidence:** 5

**Summary:**

This paper proposed a novel diffusion-based method Motion-Catcher, for multi-sequence video generation. The framework consists of a motion capture module, as well as a dynamic content prior module towards addressing the issues of motion inconsistency and content degradation. Experiments show that Motion-Catcher outperforms SoTA on visual quality and spatio-temporal consistency

**Strengths:**

1. The proposed method seems to be simple and effective for image-to-video generation.
2. Applying optical-flow in video generation seems to be reasonable and is able to provide more stable motion guidance.

**Weaknesses:**

1. Writing needs to be improved. It is not easy to follow the entire proposed idea.
2. "We present an efficient method ...", I would like to see more discussion on efficiency in the proposed method. I didn't find any comparison in the experiments section.
3. "... as plug-and-play components in different video diffusion generation models", I didn't find any experiments to demonstrate this, could authors provide results on applying proposed method on other video diffusion models to demostrate the generalizability?
4. In Figure 1, it is unclear how "reference image" and zero optical flow work. It seems that the figure lacks illustration of this part.
5. What is Motion-catcher Net and how does it function? I didn't find a detailed introduction on this network.
6. Lacking discussion and comparison with a previous method SEINE [1] in Table 1 and related work.
7. The training part is a bit vague. Does SVD need to be fined-trained? How long the entire training process take?
8. To generate a 28-frame video, what will it take? What is the maximum video-length the proposed method could generate?
9. I noticed in SM, most of the demo videos only contain large camera motion. I am curious whether the proposed method is effective for local motion such as human or animal action? I expect more diverse examples could be provided.
10. What are the limitations of the proposed method?
11. Typo: L440, Figure. 6 -> Figure 6, L485, Figure. 7 -> Figure 7
12. Typo: L449, we -> We

**Questions:**

see weaknesses

---

### Note · Authors · 2024-11-19

I have read and agree with the venue's withdrawal policy on behalf of myself and my co-authors.